# Combining Asian and European genome-wide association studies of colorectal cancer improves risk prediction across racial and ethnic populations

Polygenic risk scores (PRS) have great potential to guide precision colorectal cancer (CRC) prevention by identifying those at higher risk to undertake targeted screening. However, current PRS using European ancestry data have sub-optimal performance in non-European ancestry populations, limiting their utility among these populations. Towards addressing this deficiency, we expand PRS development for CRC by incorporating Asian ancestry data (21,731 cases; 47,444 controls) into European ancestry training datasets (78,473 cases; 107,143 controls). The AUC estimates (95% CI) of PRS are 0.63(0.62-0.64), 0.59(0.57-0.61), 0.62(0.60-0.63), and 0.65(0.63-0.66) in independent datasets including 1681-3651 cases and 8696-115,105 controls of Asian, Black/African American, Latinx/Hispanic, and non-Hispanic White, respectively. They are significantly better than the European-centric PRS in all four major US racial and ethnic groups (p-values < 0.05). Further inclusion of non-European ancestry populations, especially Black/African American and Latinx/Hispanic, is needed to improve the risk prediction and enhance equity in applying PRS in clinical practice.

Colorectal cancer (CRC) is a leading cause of cancer death, yet it is among the most preventable cancers via screening[1]. Together with the detection of CRC at early stages, which dramatically improves prognosis, optimal screening has the potential for a major impact on CRC mortality. However, current screening programs are primarily age and family-history based and more refinement through risk-based screening recommendations could be instrumental in improving their effectiveness.

Genetics plays a key role in the CRC development and, as for most cancers and other common diseases, the risk is polygenic[2]. As such, we can utilize the polygenic risk structure to develop a polygenic risk score (PRS) to quantify an individual's inherited risk of developing CRC. As the predictive performance improves, a PRS can become clinically useful as a risk stratification tool for targeted screening and chemoprevention. However, PRS built based on European ancestry data have sub-optimal performance in other ancestral populations[3] because of differential linkage disequilibrium (LD) patterns and allele frequencies across racial and ethnic groups for disease risk variants of CRC[4–9]. The poor transferability of PRS across racial and ethnic groups has raised concern regarding whether its application in clinical practice may exacerbate existing health disparities[7]. As a result, there is a need to improve the accuracy of polygenic prediction across different racial and ethnic groups to maximize the clinical and public-health translational potential of PRS and enhance equity in precision medicine.

Developing ancestry-specific PRS requires sufficient sample sizes for each ancestral group; however, the sample sizes for non-European ancestry groups, while increasing, remain only a fraction of the sample size for European ancestry. Existing studies suggest that leveraging information from other ancestries can improve ancestry-specific

✉e-mail: upeters@fredhutch.org; lih@fredhutch.org

PRS[10,11]. As an alternative to developing ancestry-specific PRS, one may develop a single cross-ancestry PRS based on meta-analysis of genome-wide association studies (GWAS) across all available ancestral groups[12–14]. To our knowledge, there is no study of PRS for non-European ancestral populations for CRC. Here we consider two different approaches to PRS development, (1) ancestry-specific PRS using PRS-CSx[15] based on ancestry-specific GWAS while leveraging cross-ancestry information and (2) single cross-ancestry Asian-European PRS using LDPred2[16] based on combined meta-analysis summary statistics and LD matrices across Asian and European ancestries. Using independent racially and ethnically diverse datasets, we evaluated the performance of these two PRS and compared them with a genome-wide PRS built using European-only GWAS data[3] and a PRS based on 204 known CRC loci[17–20]. To facilitate understanding of its clinical utility, we used decision-curve analyses[21] to assess the standardized net benefit for the model based on family-history and PRS and compared to the family-history-only model, as the latter is currently used to decide at what age screening starts.

## Results

For developing PRSs, we used GWAS summary statistics of 1,020,293 SNPs based on 21,731 cases and 47,444 controls of Asian and 78,473 cases and 107,143 controls of European ancestries. We evaluated the performance of the PRS in independent validation individual-level data sets including 12,025 Asian (2420 cases; 9605 controls), 13,823 Black/African-American (1954 cases; 11,869 controls), 10,378 Latinx/Hispanic (1682 cases; 8696 controls) and 118,756 non-Hispanic White (3651 cases; 115,105 controls) participants. More details about study participant characteristics for training and validation data sets are included in Table 1, Supplementary Data 1, and Supplemental Material and Methods.

### Discriminatory accuracy of Asian-European PRS

The single cross-ancestry Asian-European PRS derived using the combined Asian-European GWAS meta-analysis summary statistics and LD matrices with LDpred2 improved the discriminatory accuracy in the Asian population compared to the European-centric PRS (AUC = 0.63 vs. 0.59, $p$-value < 4.5e−09, Table 2). It also improved the AUC significantly in the non-Hispanic White population (AUC = 0.65 vs. 0.63, $p$-value = 6.0e−03). Despite lack of Black/African American and Hispanic individuals in deriving the PRS, the Asian-European PRS improved the AUC for Black/African American (AUC = 0.59 vs. 0.58, $p$-value = 0.05) and Hispanic individuals (AUC = 0.62 vs. 0.59, $p$-value = 5.0e−03). The Asian-European PRS improved the AUC in all racial and ethnic groups compared to the known-loci PRS (all $p$-values < 0.05).

The ancestry-specific PRS derived using PRS-CSx improved the discriminatory accuracy in the Asian population compared to the European-centric PRS (AUC = 0.64 vs. 0.59), though not statistically significant with $p$-value 0.06 (Table 2). The AUC for the ancestry-specific non-Hispanic White-specific PRS was also not statistically different from the European-centric PRS ($p$-value = 0.15) in the non-Hispanic White population; however, it was significantly higher than the known-loci PRS ($p$-value = 1.8e−05). The ancestry-specific PRS-CSx is not relevant for Black/African American and Hispanic groups, because there were no GWAS for these groups included in the training datasets.

There was little variation in AUC estimates across studies (Supplemental Table 1). Among these two approaches, the Asian-European PRS using the combined Asian-European summary statistics in LDpred2 had greater discriminatory accuracy than the ancestry-specific non-Hispanic White-specific PRS from PRS-CSx with $p$-value = 3.0e−03. However, we did not observe statistically significant differences in Asian individuals ($p$-value = 0.75). Taken together, the single cross-ancestry Asian-European PRS using LDpred2 performs among the best in terms of AUC but with much narrower confidence intervals; hereafter we focus only on the single cross-ancestry Asian-European PRS. The ROC curves for the cross ancestry Asian-European PRS showed a similar pattern to the AUC for Asian, Black/African American, Hispanic, and non-Hispanic White participants (Supplemental Fig. 1).

### PRS distribution across racial and ethnic groups

As expected, the PRS distributions varied across the racial and ethnic groups (Fig. 1A and Supplemental Fig. 2). After trans-ancestry correction, the PRS distributions largely overlapped except for the MG-JPN study (Fig. 1B and Supplemental Fig. 3). This may be due to the use of the imputation reference panel of only Asian individuals from the 1000 Genomes Projects for MG-JPN; this differs from all other studies, which used all 1000 Genome Project samples in the reference panel. We thus performed an additional mean adjustment to the PRS for the MG-JPN study. After this adjustment, all PRS distributions overlapped (Fig. 1C).

Cases had higher mean PRS than controls across all racial and ethnic groups (Supplemental Fig. 4). The OR estimates per SD of PRS (95% CI) were 1.64 (1.55–1.74), 1.39 (1.31–1.47), 1.62 (1.51–1.73) and 1.67 (1.60–1.75) for Asian, Black/African American, Latinx/Hispanic, and non-Hispanic White participants, respectively, with $p$-value < 2.0e−18 for all four groups (Fig. 1D and Table 3).

Compared to the mean risk, the relative risks of PRS at any given percentile were similar for all racial and ethnic groups except for Black/African American participants for whom it was attenuated (Fig. 2). The relative risk at the 90th percentile of the PRS distribution compared to mean was 1.67, 1.44, 1.65, and 1.69 for Asian, Black/African American, Latinx/African American, and non-Hispanic participants, respectively.

The model-based relative risk was calibrated well across the PRS range in all racial and ethnic groups (Fig. 3).

## Table 1 | Characteristics of the validation studies

| Study | Racial and Ethnic Group | Total N | CRC No. (%) | Mean age (range) | Female No. (%) | Family-history | | |
|---|---|---|---|---|---|---|---|---|
| | | | | | | Yes No. (%) | No No. (%) | Missing No. (%) |
| GERA | Asian | 7370 | 96 (1.0%) | 64 (19–95) | 4152 (56.3%) | 643 (8.7%) | 6727 (91.3%) | 0 (0%) |
| | Black or African American | 3159 | 56 (2.0%) | 66 (20–95) | 1811 (57.3%) | 313 (9.9%) | 2846 (90.0%) | 0 (0%) |
| | Latinx or Hispanic | 6660 | 70 (1.0%) | 63 (19-95) | 4081 (61.3%) | 543 (8.2%) | 6117 (91.8%) | 0 (0%) |
| | non-Hispanic White | 77,012 | 1401 (1.8%) | 70 (19–95) | 44,125 (57.2%) | 7423 (9.6%) | 69,589 (90.4%) | 0 (0%) |
| MG | Japanese | 4655 | 2324 (50.0%) | 65 (20–90) | 2007 (43.1%) | 407 (8.7%) | 1410 (30.0%) | 2838 (61.0%) |
| | Black or African American | 6597 | 1856 (28.1%) | 66 (20–91) | 2581 (39.1%) | 721 (11.0%) | 934 (14.0%) | 4942 (74.9%) |
| Hispanic GWAS | Latinx or Hispanic | 3717 | 1611 (43.3%) | 65 (21–90) | 1790 (48.1%) | 260 (7.0%) | 2949 (79.3%) | 508 (13.7%) |
| CPSII | non-Hispanic White | 1712 | 804 (46.9%) | 71 (54–90) | 769 (44.9%) | 204 (11.9%) | 1417 (82.8%) | 91 (5.3%) |
| BCC | non-Hispanic White | 1818 | 873 (48.0%) | 57 (8-99) | 627 (34.4%) | 0 | 0 | 1818 (100.0%) |
| eMERGE | Black or African American | 4067 | 42 (1.0%) | 52 (18–90) | 2946 (72.0%) | 105 (2.6%) | 3705 (91.0%) | 257 (6.3%) |
| | non-Hispanic White | 38,214 | 573 (1.5%) | 65 (18–90) | 20,543 (54.0%) | 965 (2.5%) | 30,157 (78.90%) | 103 (0.3%) |

**Table 2 | AUC estimates (95% confidence interval) for European-centric PRS, known loci PRS, PRS-CSx and LDPred2**

| Race and Ethnicity | Cases/ controls | European-centric PRS | Known Loci PRS | PRS-CSx | LDPred2 |
|---|---|---|---|---|---|
| Asian | 2420/9605 | 0.59 (0.57–0.60) | 0.60 (0.59-0.62) <br> *p*-val*: 0.24 | 0.64(0.58–0.69) <br> *p*-val*: 0.06 <br> *p*-val⁺: 0.32 | 0.63 (0.62–0.64) <br> *p*-val*: 4.5e−9 <br> *p*-val⁺: 1.6e−6 <br> *p*-val**: 0.75 |
| Black or African American | 1954/11,869 | 0.58 (0.56–0.59) | 0.58 (0.56–0.59) <br> *p*-val*: 0.92 | | 0.59 (0.57–0.61) <br> *p*-val*: 0.05 <br> *p*-val⁺: 0.01 |
| Latinx or Hispanic | 1681/8696 | 0.59 (0.57–0.61) | 0.59 (0.57–0.60) <br> *p*-val*: 0.76 | | 0.62 (0.60–0.63) <br> *p*-val*: 5.0e−3 <br> *p*-val⁺: 1.0e−3 |
| Non-Hispanic White | 3651/115,105 | 0.63 (0.62–0.65) | 0.61 (0.60–0.62) <br> *p*-val*: 9.0e−4 | 0.64 (0.62-0.65) <br> *p*-val*: 0.15 <br> *p*-val⁺: 1.8e−5 | 0.65 (0.64–0.66) <br> *p*-val*: 6.0e−3 <br> *p*-val⁺: 1.1e−14 <br> *p*-val**: 3.0e−3 |

The *p*-values are two-sided and calculated based on 500 bootstrapping samples.
*p*-value comparison of PRS with the European-centric PRS.
⁺*p*-value comparison of PRS with the known-loci PRS.
**p*-value comparison of PRS with the PRS-CSx.
**All AUC estimates were adjusted for age, sex, and top 4 principal components.

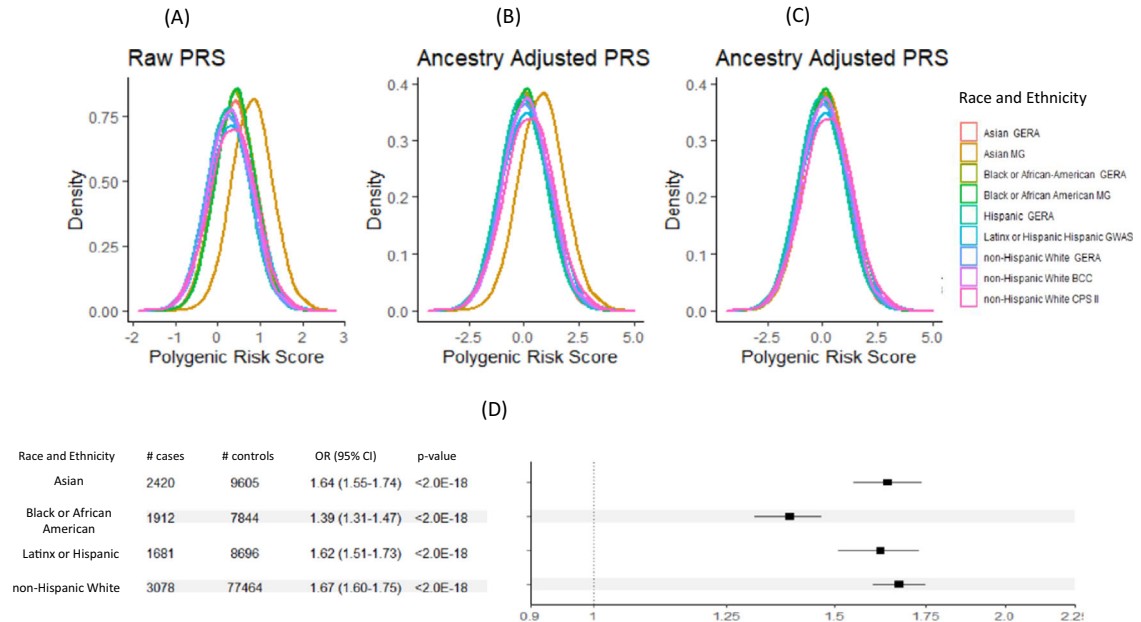

**Fig. 1 | Distribution of PRS. A** PRS distributions varied across racial/ethnic groups, **B** PRS distribution after ancestry adjustment, **C** Additional mean adjustment for the Asian MG (Minor GWAS Japanese Study) study that has a different imputation panel, and **D** forest plot by racial and ethnic group for OR estimates +/−1.96 standard error of PRS per SD using *N* = 120,25; 9756; 10,377 and 80,542 of unrelated samples of Asian, Black or African American (AA), Hispanic and non-Hispanic White, respectively. The *p*-values in the table are two-sided. PRS is based on single cross-ancestry Asian-European PRS.

## Odds ratios (ORs) for PRS stratified by family-history and age

Across all racial and ethnic groups, the ORs for the PRS were higher in those without a family-history than those with a family-history with *p*-values 0.21, 0.01, 3.0e−3, and 0.11 for Asian, Black/African American, Latinx/Hispanic, and non-Hispanic White participants respectively (Table 3). The estimates were consistent across studies (Supplemental Table 2).

The strength of association estimates for PRS in relation to CRC decreased over strata of increased age in each racial and ethnic group with trend test *p*-values of 0.07, 0.11, 2.8e−4, and 1.2e−03 for Asian, Black/African American, Latinx/Hispanic, and non-Hispanic White, participants, respectively. The ORs, 95% CI and trend *p*-value for each racial and ethnic group are given in Table 3. The estimates were consistent across studies (Supplemental Table 2).

## Clinical utility for model based on PRS and family-history

We calculated the standardized net benefit (sNB) to assess the clinical utility of using a model based on PRS and family-history to recommend an intervention (such as screening) for participants <50 years of age. We used the average 10-year risk of developing CRC at age 45 as the risk threshold, because the current CRC-screening guidelines recommend that an average-risk individual start screening at age 45 years old. Using the GERA cohort, we estimated the 10-year risk to be 0.29% across all racial and ethnic groups. At this risk threshold, the risk model based on PRS, and family-history achieved 37.3% (95% CI: 23.8%–50.8%) of the maximum possible achievable utility. This was greater than the model based on family-history alone (sNB = 21.7%, 95% CI: 12.4%–33%, *p*-value 0.02) and hypothetically intervening on all or no people

**Table 3 | Odds ratios (OR), 95% confidence interval (95% CI) and two-sided *p*-values for PRS per SD for all and stratified by family-history and age**

| | Asian OR (95% CI) *p*-value | Black or African American OR (95% CI) *p*-value | Latinx or Hispanic OR (95% CI) *p*-value | Non- Hispanic White OR (95% CI) *p*-value |
|---|---|---|---|---|
| **PRS per SD** | 1.64 (1.55–1.74) <2.00e–18 | 1.39 (1.31–1.47) <2.00e–18 | 1.62 (1.51–1.73) <2.00e–18 | 1.67 (1.60–1.75) <2.00e–18 |
| **Family-history** | | | | |
| No | 1.62 (1.46–1.80) <2.0e–18 | 1.53 (1.26–1.85) 1.21e–5 | 1.64 (1.52–1.72) <2.0e–18 | 1.67 (1.59–1.76) <2.0e–18 |
| Yes | 1.45 (1.19–1.76) 1.9e–4 | 1.22 (1.03–1.44) 1.2e–2 | 1.15 (0.91–1.46) 0.245 | 1.51 (1.36–1.67) 9.10e–15 |
| **Age** | | | | |
| <50 | 1.88 (1.50–2.35) 4.35e–8 | 1.51 (1.17–1.94) 1.33e–3 | 1.17 (0.68–2.00) 5.7e–1 | 1.85 (1.48–2.31) 5.31e–8 |
| 50–60 | 1.85 (1.62–2.12) <2.0e–18 | 1.53 (1.36–1.72) 8.56e–13 | 2.15 (1.77–2.562) 1.29e–14 | 1.75 (1567–1.96) <2.0e–18 |
| 60–70 | 1.58 (1.43–1.74) <2.0e–18 | 1.41 (1.28–1.55) 1.49e–12 | 1.58 (1.42–1.76) <2.0e–18 | 1.88 (1.73–2.04) <2.0e–18 |
| 70–80 | 1.57 (1.41–1.75) 1.33e–15 | 1.31 (1.17– 1.45) 1.39e–6 | 1.47 (1.30–1.67) 1.48e–9 | 1.61 (1.49–1.74) <2.0e–18 |
| >80 | 1.65 (1.28–2.13) 1.21e–4 | 1.32 (1.03– 1.69) 3.07e–2 | 1.58 (1.211–2.05) 6.903–4 | 1.43 (1.30– 1.58) 2.12e–13 |
| **Trend *p*-value** (Age) | 7.0e–2 | 0.11 | 2.8e–4 | 1.2e–3 |

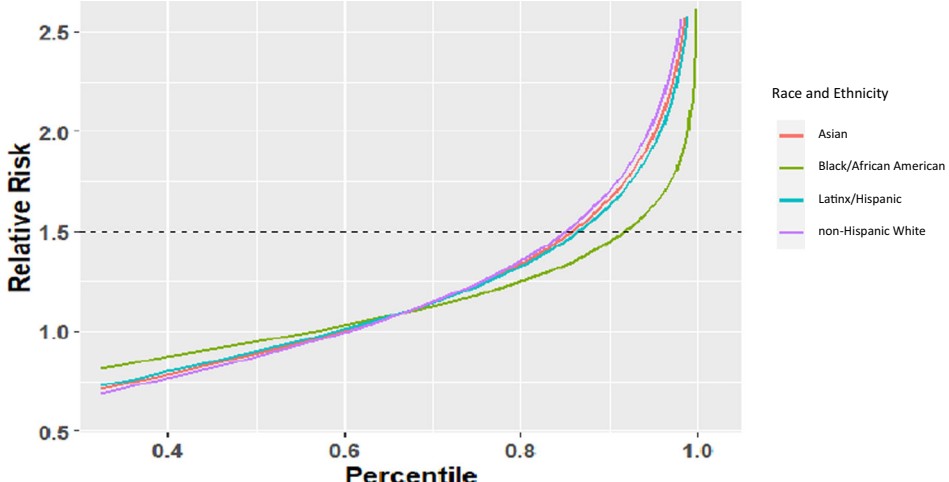

**Fig. 2 | Relative Risk Estimation.** The relative risk of individuals at different percentiles of the single cross-ancestral Asian-European PRS compared to a population average odds ratio, stratified by race and ethnicity.

(Fig. 4a), a pattern that generally holds for each racial and ethnic group (Supplemental Fig. 5).

We observed a similar pattern for participants between the ages of 50 and 60 years (Supplemental Fig. 6). We also used the 10-year risk 0.39% at age 50 and 0.49% at 55 years as the risk thresholds. The risk model based on PRS, and family history achieved greater sNB (sNB = 24.8% and 21.6%, respectively) than the model based on family history-alone (sNB = 19.3% and 15.9%, respectively).

At the risk threshold 0.29%, in GERA cohort, for the model based on family history and PRS, the true-positive and false-positive rates were 70% and 37%, respectively, whereas, for the model based on family history only, the true-positive and false-positive rates were 31% and 10%, respectively (Fig. 4b). About 8472 of 22,628 individuals with age 40–49 were deemed to be at high risk based on our model of family history and PRS. Among these, 99 developed CRC in the next 10 years. For this age group, a total of 149 individuals developed CRC. Whereas, for the model based on family history only, at the same risk threshold, about 2357 would be deemed at high risk, and 37 developed CRC. (Fig. 4c, d).

Table 4 provides more detailed results of the net benefit (NB) analysis for our proposed family history and PRS-based model and the family history-based model compared to treat all for risk thresholds (%) from 0 to 0.32%, where NB for treat all becomes negative. Using the same risk threshold 0.29% as in the previous example, the NB of our model is 0.11%. This can be interpreted as that compared with assuming that all individuals do not have intervention, our model with 0.11% NB leads to the equivalent of a net 11 true-positives per 10,000 individuals without an increase in the number of false-positives. Moreover, the net benefit for the model was 0.08% greater than assuming all individuals had intervention and 0.04% greater than family history-based model. We also calculated the reduction in the number of false positives per 100 patients as[22]. There were 30 fewer false-positives per 100 individuals for our models whereas there were only 15 fewer false-positives for the family history-based model.

In addition, we estimated the number of unnecessary interventions avoided for individuals with age 40–49 years old, as shown in Supplemental Fig. 7 and Table 5. Continuing using the 0.29% threshold as an example, risk stratification based on the family history and PRS

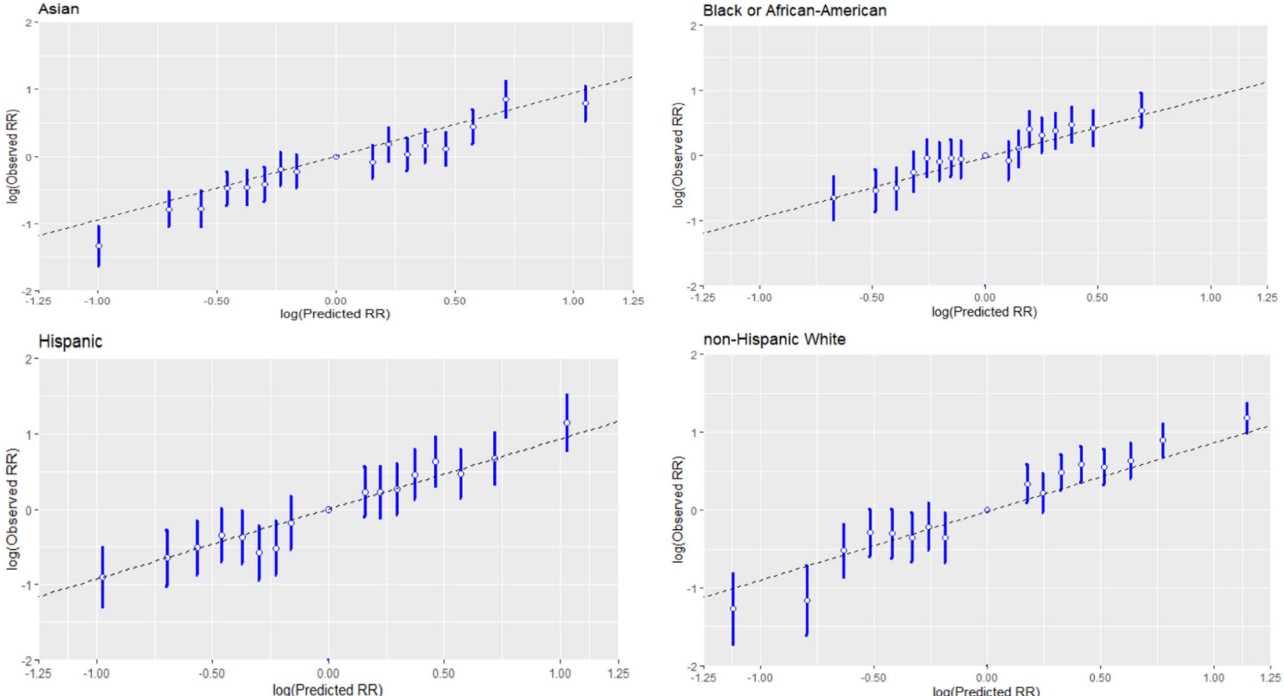

**Fig. 3 | Relative Risk Calibration of PRS.** The relative risk calibration of PRS, stratified by race and ethnicity, using N = 120,25; 9756; 10,377 and 80,542 of unrelated samples of Asian, Black or AA, Hispanic and non-Hispanic White, respectively. The *x*-axis is the log-transformed predicted RR values and the *y*-axis is the log-transformed observed RR +/− 1.96 standard error with the middle bin (40–60) as the reference group.

would avoid 17 more interventions per 100 individuals, compared with the model based on family history, which would avoid 13 interventions per 100 individuals compared to intervening all.

### Assessing CRC probabilities for PRS

We estimated age-specific probabilities for developing CRC by age 80, stratified by family history status, and by quantiles of PRS top 5%, top 25%, 25%−75%, bottom 25% and bottom 5%, for different racial/ethnic groups of GERA participants. There was clear separation between those who were in bottom and top PRS quantiles across ancestral groups, except for the African American group where the separation is less obvious due to the lower performance and very limited number of CRC cases in this group. The probabilities of developing CRC by age 70 for top 5% of PRS ranged from 2.2 to 4.7%, across the four different racial and ethnic groups. In comparison, the probabilities of developing CRC for those who had the positive family history were 1.9–5% (Supplemental Figs. 8 and 9).

### Discussion

Using large-scale Asian and European GWAS data, we demonstrate that combining Asian and European summary statistics in deriving PRS led to statistically significant improvement in discriminatory accuracy across Asian, Black/African American, Latinx/Hispanic and non-Hispanic White groups, although the improvement was less marked in Latinx/Hispanic and Black/African American participants. We further show that across all groups, the PRS has stronger associations with CRC-risk in younger individuals and in those without a family-history of CRC, which will likely increase the possible clinical utility of the PRS given the rising young-onset CRC incidence rates in recent decades, mostly in individuals without a known family-history. This is supported by our decision-curve analysis demonstrating that adding PRS improves the maximum achievable clinical utility over the model based on family-history only for ages 40–60 years.

A challenging factor of moving PRS to clinical implementation is ensuring that the PRS is equally applicable to individuals across all racial and ethnic groups to prevent an increase in health disparities. Relevant to this objective, we evaluated two broad categories of approaches (ancestry-specific PRS while leveraging cross ancestry information and single cross-ancestry PRS based on the combined cross-ancestry GWAS) for improving the prediction in under-represented groups, and our observation of the performance of these approaches could be generalized to other traits besides CRC. We found that both approaches performed similarly in Asian and non-Hispanic European individuals. Further, the cross-ancestry Asian-European PRS also improved risk prediction performance in Hispanic individuals and, to a smaller extent, in Black/African American individuals. We also show that we can correct this raw PRS for genetic ancestry and create a common distribution that can be used across racial and ethnic groups, avoiding the potential difficulty of using ancestry-specific PRS in admixed populations. Accordingly, our cross-ancestry Asian-European PRS has the potential to reduce health disparities between non-European ancestry populations and the European ancestry population.

As there is growing interest in clinical use of PRS, it is important to point out that the purpose of PRS is not to identify CRC, but rather stratify individuals into different risk strata for which different levels of cancer preventive interventions may be devised.[23,24] Their performance should thus be compared with risk factors currently used for risk stratification such as family-history in terms of cost effectiveness. In this paper, we performed a decision-curve analysis that has been used in cancer research for assessing the potential population impact of incorporating a risk prediction model into clinical practice[22,25,26]. The risk model that incorporates both the PRS and family-history achieves 37.3% of the maximum possible achievable utility for those 40–49 years old, significantly greater than 21.7% under the family-history-only model. Recently the US Preventive Services Task Force recommended lowering the age at screening initiation to 45 years for individuals at average risk[27]. However, given the substantial burden of additional approximately 22 million people becoming eligible for screening and the fact that CRC remains a rare event in younger individuals, there has

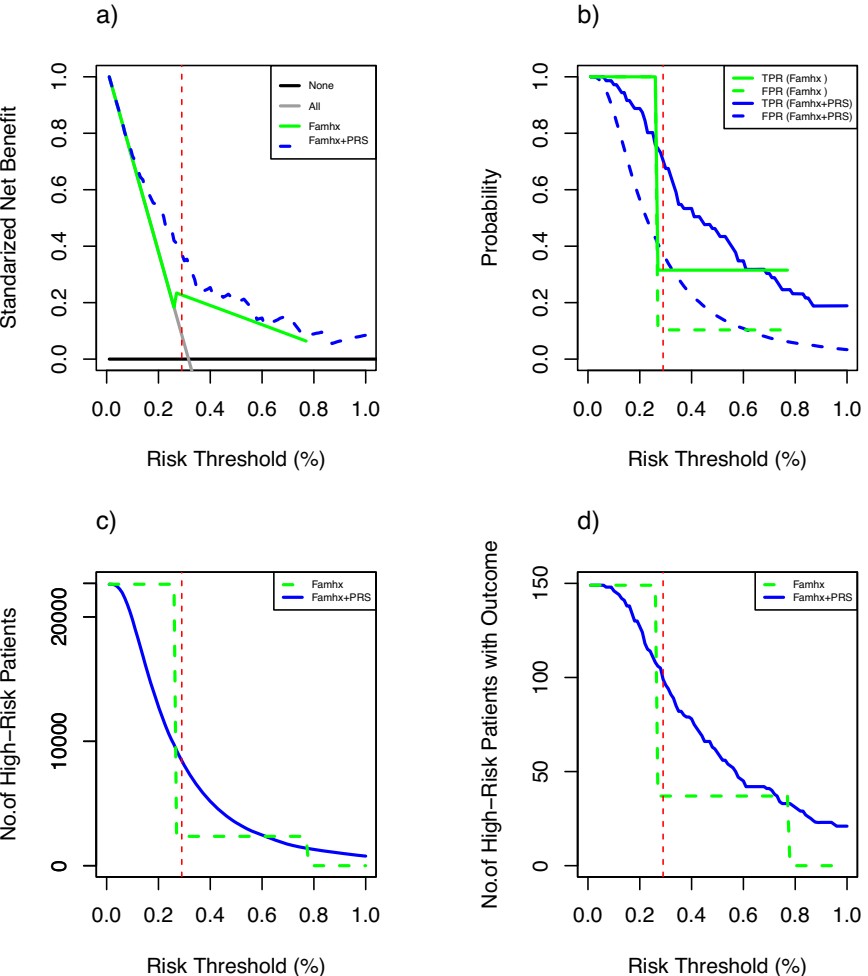

**Fig. 4 | Standardized net benefit analysis. a** Standardized net benefit for none, all, family history (FamHx) model, and FamHx+PRS model. For the FamHx and Famhx+PRS models **b** true- and false-positive rates, **c** number of high-risk, and **d** number of high risk participants developed CRC at different risk thresholds, in 22,628 participants aged 40–49 from the GERA cohort.

been critique of the universal change to the initial screening age that, instead, emphasizes the importance of targeted screening based on an individual's risk factors[28–30]. The results from the decision-curve analysis suggest that there is clinical utility to adding a PRS to the family-history-only model in risk stratification for CRC prevention. In decision curve analysis, we assumed the decision in question was whether an individual in the general population should undergo intervention (e.g., colonoscopy procedure), based on their risk. Overall, the model with the highest (standardized) net benefit is considered the "best" strategy in decision curve analysis. However, as argued in Kerr et al.[21], decision curves cannot be used to choose a risk threshold, but it summarizes the costs and benefits of intervention of the risk model at different risk threshold. To fully evaluate the effectiveness of including PRS as part of risk stratification, a full decision analytic modeling that incorporates other aspects such as different screening methods, implementation factors, behavioral factors, and corresponding costs are warranted[31].

Recent efforts[32,33] in clinical implementation of PRS shows the potential of PRS to effectively stratify the risk of diseases development and guide screening. BOADICEA v5 (as implemented in the CanRisk tool)[32] already implements a 313-variant PRS of breast cancer and currently supports hundreds of thousands of women, doctors, and genetic counselors annually in >90 countries making treatment decisions. PRS-guided mammographic screening is also being tested in the WISDOM and PERSPECTIVE I&I studies[33]. GenoVA Study[34] is a clinical

trial in which patients and their primary care physicians receive a clinical PRS laboratory report on five diseases including CRC. MyOme implements a cross-ancestry risk score for breast cancer risk stratification[35]. As CRC has an effective screening intervention, it would be of great interest to explore implementation of PRS for guiding personal screening recommendations.

This study has several strengths. We brought together most of the globally available GWAS of CRC for Asian and European ancestry populations as our training data, which is an important factor for the improved performance of the proposed PRS. Further, we used multiple independent evaluation data sets that were not part of our training data nor GWAS discovery, providing an unbiased evaluation of the developed models. Moreover, the single cross-ancestral PRS derived in this study makes it easy to implement in any admixed population.

The results of this investigation should be interpreted in the context of its limitations. The discriminatory accuracy remains lower in Latinx/Hispanic and particularly in Black/African American individuals due to their limited sample sizes in training data. Future studies more inclusive of these individuals are warranted for deriving PRS to enhance the discriminatory accuracy. Furthermore, we have not been able to evaluate the performance of these models in other racial and ethnic groups, including Alaskan Native, Native American and Pacific Islander individuals. Lastly, we expect to further improve risk prediction by combining the PRS with non-genetic risk factors such as obesity, diet, and aspirin use, as previously shown[24,36].

**Table 4 | Net benefit (NB) of intervention (e.g., screening) for 22,628 participants aged 40–49 from the GERA cohort, according to the proposed family history (FamHx) + PRS model and the FamHx only model for a given risk threshold**

| Risk threshold (%) | Treat All | NB | | Advantage of model compared to treat all | | | | |
| | | FamHx Model | FamHx + PRS Model | FamHx Model | | | FamHx + PRS Model | |
| | | | | Net benefit | Reduction of false positive per 100 | | Net benefit | Reduction of false positive per 100 |
|---|---|---|---|---|---|---|---|---|
| 0.04 | 0.002763 | 0.002763 | 0.002766 | 0 | 0 | | 3.05E−06 | 1 |
| 0.08 | 0.002364 | 0.002364 | 0.002378 | 0 | 0 | | 1.40E−05 | 2 |
| 0.09 | 0.002264 | 0.002264 | 0.002307 | 0 | 0 | | 4.33E−05 | 5 |
| 0.11 | 0.002064 | 0.002064 | 0.002151 | 0 | 0 | | 8.66E−05 | 8 |
| 0.13 | 0.001864 | 0.001864 | 0.001976 | 0 | 0 | | 0.000112 | 9 |
| 0.14 | 0.001764 | 0.001764 | 0.001945 | 0 | 0 | | 0.00018 | 13 |
| 0.16 | 0.001564 | 0.001564 | 0.001809 | 0 | 0 | | 0.000245 | 15 |
| 0.17 | 0.001464 | 0.001464 | 0.001751 | 0 | 0 | | 0.000286 | 17 |
| 0.19 | 0.001264 | 0.001264 | 0.001683 | 0 | 0 | | 0.000419 | 22 |
| 0.20 | 0.001164 | 0.001164 | 0.001678 | 0 | 0 | | 0.000513 | 26 |
| 0.22 | 0.000964 | 0.000964 | 0.001545 | 0 | 0 | | 0.000581 | 26 |
| 0.23 | 0.000864 | 0.000864 | 0.00141 | 0 | 0 | | 0.000547 | 24 |
| 0.25 | 0.000663 | 0.000663 | 0.00142 | 0 | 0 | | 0.000757 | 30 |
| 0.26 | 0.000563 | 0.000563 | 0.001285 | 0 | 0 | | 0.000722 | 28 |
| 0.28 | 0.000363 | 0.000707 | 0.001222 | 0.000344 | 12 | | 0.000859 | 31 |
| 0.29 | 0.000263 | 0.000696 | 0.001142 | 0.000434 | 15 | | 0.00088 | 30 |
| 0.31 | 6.20E−05 | 0.000676 | 0.001084 | 0.000613 | 20 | | 0.001022 | 33 |
| 0.32 | −3.83E−05 | 0.000665 | 0.001006 | 0.000703 | 22 | | 0.001045 | 33 |

Advances in PRS development have promoted the use of PRS-enhanced models to determine and stratify disease risk, which could improve disease prevention and management through screening and early detection. Our cross-ancestry Asian-European PRS, built upon data on both Asian and European ancestry individuals, improves the PRS performance in Asian, Black/African, and Latinx/Hispanic individuals considerably. Combining PRS and other CRC-associated risk factors such as lifestyle/environmental risk factors and high penetrance genes will likely further improve the prediction performance[36]. We anticipate that the continuous expansion of PRS development and validation to include more diverse populations and prospective evaluation of PRS-enhanced risk prediction model in clinical trials along with decreasing genotyping cost and adaptation of health care systems to accommodate genetic data and prediction algorithm will bring closer the implementation of PRS in clinical practice.

## Methods

### Training data sets

To develop polygenic risk scores (PRS) across population, we used the genome-wide association study (GWAS) summary statistics of 1,020,293 SNPs based on 78,473 cases and 107,143 controls of European (EUR) and 21,731 cases and 47,444 controls of Asian ancestries from GWAS catalog under accession code GCST90129505 (Supplementary Data 1)[17–19]. For this we group participants into analytical units by study or genotyping platform as consistent with the original reports[17–20,37,38]. Ancestry was determined by the genetic principal component analysis. Studies that contributed to more than one prior genome-wide association analyses were analyzed only once. In total, there were 31 analytical units (17 from EUR descent populations and 14 from Asian descent populations), totaling 100,204 CRC cases and 154,587 controls. Comprehensive details on the participants, genotyping and standard quality control (QC) procedures are summarized in Supplementary Data 1. All study protocols were approved by the relevant Institutional Review Boards, and informed consent was obtained from all study participants in accordance with the Helsinki accord.

### Independent validation data sets

We evaluated the performance of each of the developed PRS in the Genetic Epidemiology Research on Adult Health and Aging Cohort (GERA) cohort; Minority GWAS Japanese study (MG-JPN)[39]; Minority GWAS African American study (MG-AA)[40]; Hispanic Colorectal Cancer Study (HCCS)[41]; Multiethnic Cohort study (MEC); Cancer Prevention Study II (CPSII)[42]; Basque-colon cohort (BCC); and Electronic Medical Records and Genomics (eMERGE) study. Racial and ethnic identification in these studies were self-reported. In total, there were 12,025 Asian (2,420 cases; 9605 controls), 13,823 Black/African-American (1954 cases; 11,869 controls), 10,378 Latinx/Hispanic (1682 cases; 8696 controls) and 118,756 non-Hispanic White (3651 cases; 115,105 controls) participants. None of these samples was included in the training data sets for model building. More details about study participant characteristics are included in Table 1.

CRC status (Yes/No) was determined from cancer-registry data. Family-history of CRC (>=1 first-degree relatives with CRC), was ascertained through baseline study questionnaire or electronic medical records at study entry.

### Approaches for deriving PRS

We compared two different approaches for PRS development using (1) ancestry-specific PRS using PRS-CSx that integrates genome-wide Asian and European summary statistics and LD matrices; (2) single cross-ancestry PRS using LDpred2 that combine genome-wide Asian and European summary statistics and a weighted LD matrix with weight defined as the proportion of participants from each ancestry in the summary statistics. Figure 5 depicts the summary of these PRS derivations.

PRS-CSx[15] derives ancestry-specific PRS while leveraging GWAS summary statistics from other ancestral groups. We first obtained ancestry-specific PRS using ancestry-specific GWAS summary statistics and LD matrix for Asian and non-Hispanic White participants based on ~1M genome-wide SNPs, respectively, while leveraging GWAS from the other ancestral group. We denoted these PRS by $PRS_{Asian}$ and $PRS_{European}$, respectively. We then improved ancestry-specific PRS by

taking a weighted sum of these PRSs to predict CRC of respective ancestral group. To derive PRS for the Asian population, we calculated a weighted sum of $PRS_{Asian}$ and $PRS_{European}$ ($\alpha_1 PRS_{European} + \beta_1 PRS_{Asian}$) and obtained $\alpha_1$ and $\beta_1$ from a logistic regression model using the MG-JPN study. Similarly, to derive PRS for the European population, we calculated a weighted sum of $PRS_{Asian}$ and $PRS_{European}$ ($\alpha_2 PRS_{European} + \beta_2 PRS_{Asian}$), where $\alpha_2$ and $\beta_2$ were obtained based on the pooled BCC and CPSII studies.

To derive the single cross-ancestry PRS using LDpred2[16], we combined the summary statistics from the Asian and European GWAS using the inverse variance weighted estimator[43] and combined the LD matrices, as the weighted sum of the Asian and European-specific LD matrices with the weights proportional to the sample sizes of the Asian and European individuals in the combined summary statistics.

We compared ancestry-specific and single cross-ancestry PRS from PRS-CSx and LDpred2 with a previously published European-centric genome-wide PRS[3] and a known-loci PRS consisting of 204 independently CRC-associated variants based on GWAS of European and Asian ancestries[17–20] (Supplementary Data 2). Our model was focused on only PRS development and did not include any lifestyle and environmental risk factors.

**Table 5 | Unnecessary interventions avoided per 100 individuals with age 40–49 for different risk thresholds, 0.29%, 0.39% and 0.49% corresponding the average 10-year risk of developing CRC at ages 45, 50 and 55 years, for the proposed family history (FamHx) + PRS model and the FamHx only model**

| Risk threshold (%) | Famhx | Famhx + PRS |
|---|---|---|
| 0.29 | 13 | 30 |
| 0.39 | 34 | 38 |
| 0.49 | 45 | 49 |

## Evaluation of model performance

We evaluated the model performance using a wide range metrics, the Area Under the Receive Operating Characteristics curve, ancestry adjustment of PRS distribution, odds ratio estimates, and relative risk calibration based on all of the validation datasets listed in Table 1. The decision curve analysis is based on the GERA study, which was the only cohort study among our independent validation datasets.

## The area under the receiver operating characteristics curve (AUC)

We evaluated the predictive performance of the PRS by the area under the receiver operating characteristics curve (AUC) in each of the racial and ethnic groups[44]. We calculated the adjusted AUC of PRS for each study using the ROCt R package[45], adjusting for covariates age, sex and four PCs. We emphasize that the AUC estimate was for PRS only and the covariates were not part of prediction along PRS. These covariates were included as potential confounders. We then combined the AUC estimates of PRS across studies for each ancestry using the inverse variance weighted estimator.

We obtained the bootstrapped-based standard error (se), 95% confidence intervals (CI) (1.96* se) and two-sided $p$-values for comparisons across various subgroups using 500 bootstrap samples.

## Ancestry adjustment of PRS distribution

As the PRS distributions were different across racial and ethnic groups due to different allele frequencies, we used a modified trans-ancestry adjustment of PRS to align the PRS distributions[46]. We used the 1000 Genome dataset to estimate the ancestry adjustment following the approach in Khera et al.[46]. Specifically, we derived principal components (PCs) based on 343,662 ancestry informative SNPs with little overlapped (0.3%) with SNPs used in PRS development. To correct for the mean and variance differences between ancestry groups, we fit two linear regression models to predict the mean and variance of PRS based on the first four PCs. To correct for the raw PRS distribution in our data set, we first calculated the PCs using the same loadings for the

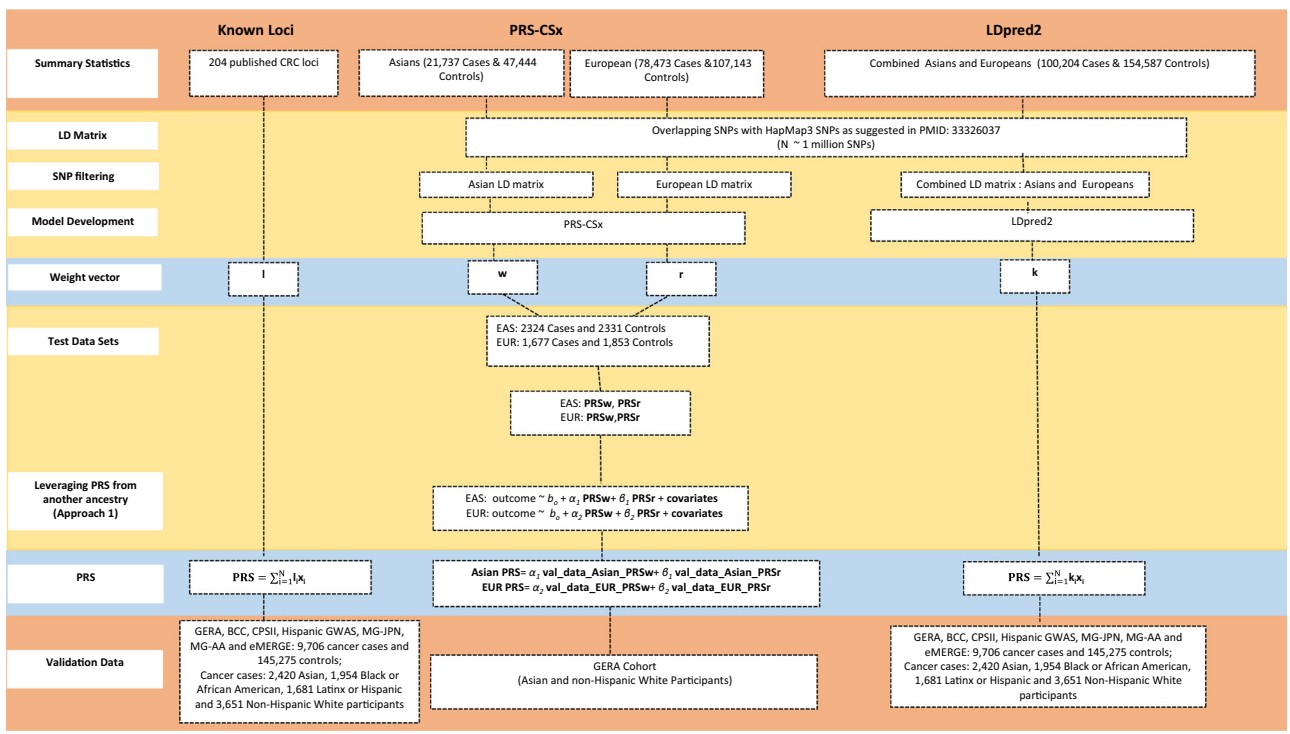

**Fig. 5 | Approaches for deriving polygenic risk scores (PRS) for colorectal cancer.** Known Loci PRS and the details of the two different approaches for deriving PRS (1) PRS-CSx PRS and (2) LDpred PRS.

top 4 PCs from the 1000 Genome data set. We then obtained the ancestry-adjusted PRS for each individual by subtracting the predicted mean based on the 4 PCs from the individual's raw PRS and then divided it by the predicted standard deviation based on the 4 PCs. Additional adjustments are needed for data sets with different imputation panels. The ancestry adjusted PRS is computed as given below:

$$PRS_{adjusted} = \frac{PRS_{sample} - (\alpha_o + \sum_{i=1}^{4} \alpha_i PC_i)}{\sqrt{\beta_o + \sum_{i=1}^{4} \beta_i PC_i}} \quad (1)$$

### Odds ratio (OR) estimates
We estimated the OR and 95% CI of CRC-risk associated per SD change in PRS by logistic regression model, overall and stratified by family history and age. For each racial and ethnic group, we estimated the AUC and OR by study and combined the estimates using the inverse variance weighted estimator. In addition, we estimated OR stratified by family history of 1st degree relative with CRC (yes, no) and age (<50, 50–59, 60–69, 70–79, and >80). All analyses were adjusted for age, sex, and top 4 principal components of ancestry.

### Relative risk calibration of PRS
We binned PRS into 5% strata and defined the reference group as PRS in the 40–60% stratum. The expected OR for a PRS stratum is the ratio of the within-stratum geometric average of individuals' model-based OR, defined as exponent of individuals' PRS times log (OR), between that stratum and the reference stratum. We estimated the observed OR estimates and its 95% CI by fitting a logistic regression model with CRC disease status as outcome and a binary variable with 1 indicating a specific stratum and 0 indicating the reference stratum, adjusting for age, sex, and first four principal components.

### Decision curve analysis
The decision-curve analysis was performed by calculating the standardized net benefit (sNB), defined as the net benefit divided by the maximum possible net benefit[21], to assess the potential clinical impact of the risk prediction models on recommended interventions (i.e., screening). For a given risk threshold, the $NB$ was defined as

$$NB = sensitivity \times p - (1 - specificity) \times (1 - p) \times w, \quad (2)$$

where $w$ was the odds at the threshold, sensitivity was the proportion of cases above the risk threshold based on the model, specificity was the proportion of controls below the risk threshold based on the model, and $p$ was the disease probability at the landmark time. As it was difficult to interpret $NB$ itself, we followed the approach proposed by Kerr et al.[21] to calculate $sNB$, i.e., dividing $NB$ by the maximum $NB$, which is achieved when $sensitivity = 1$ and $specificity = 1$. Hence, the sNB was equal to

$$sNB = sensitivity - (1 - specificity) \times \frac{(1 - p)}{p} \times w, \quad (3)$$

It provided some sense of magnitude of sNB on a percent scale and was interpreted as the relative utility that has maximum value of 1. For example, if $sNB = 0.4$, it means that the risk model achieves 40% of the maximum possible achievable utility.

To calculate the NB in the presence of competing risks[47], we denote $rt$ be the risk threshold and $I(t)$ the cumulative incidence of developing CRC for an individual by time $t$ in the presence of competing risks, here, death. Further, we define $z = 1$ to indicate that an individual is at high risk if their predicted t-year risk from the model is greater than or equal to $rt$ and $z = 0$ otherwise. We chose the landmark time $t = 10$ years. At each rt, we calculated the number of true and false positives, $TP_{rt}$ and $FP_{rt}$, by

$$TP_{rt} = I(t|z = 1) \times P(z = 1) \times N \quad (4)$$

$$FP_{rt} = \{1 - I(t|z = 1)\} \times P(z = 1) \times N \quad (5)$$

where N is the total number of participants. The true-positive rate was then calculated as $TP_{rt}/TP_{rt=0}$ and the false-positive rate was calculated as $FP_{rt}/FP_{rt=0}$. We also calculated the reduction in the number of false positives per 100 patients as[22]: *(net benefit of the model − net benefit of treat all)/{rt/(1− rt)) × 100*. We compared the model based on PRS and family history with the model based on family history alone, as well as two hypothetical extreme scenarios: intervention (e.g., screening) for all and intervention for none. We calculated the sNB under the competing risks framework[48], where the observational time is the minimum of time to CRC, time to death, and time at last observation, and the disease status is 1 if the study participant had CRC, 2 if the participant died (competing event), and 0 otherwise. We plotted decision-curves of sNB at the 10-year landmark time vs. risk threshold for age at study entry 40–49 and 50–59 years old, because average-risk individuals in these age groups are recommended to start CRC screening.

We performed the analyses using R version 4.0.0[22,45,49–51]. A two-sided *p*-value < 0.05 is considered statistically significant.

### Reporting summary
Further information on research design is available in the Nature Portfolio Reporting Summary linked to this article.

## Data availability
The Summary-level data for the full set of Asian and European GWASs used in this study are available in the GWAS catalog under accession code GCST90129505. Genotype data of GERA participants who consented to having their data shared with dbGaP are available from dbGaP under accession phs000674.v2.p2. The complete GERA data are available upon successful application to the KP Research Bank. Genotype data of eMERGE participants are available from dbGaP under the accession number phs001616.v1.p1. For individual-level data, MEC, CCFR, The MD Anderson Colorectal Cancer Case Control Study, HCCS are deposited in dbGaP (phs000220.v2.p2, phs002733.v1.p1, phs002691.v1.p1, phs001193.v1.p1) and PLCO (phs001286.v3.p2). SCCS and CanCORS data can be accessed via websites http://ors.southerncommunitystudy.org and http://outcomes.cancer.gov/cancors/. For the remaining studies please contact the corresponding PIs: CR2&3 (Loic Le Marchand at loic@cc.hawaii.edu), Fukuoka, (Loic Le Marchand at loic@cc.hawaii.edu), Nagano, JPHC(Motoki Iwasaki at moiwasak@ncc.go.jp), UNC-Rectal (Temitope Keku at temitope_keku@med.unc.edu) and Basque Study(Prof Luis Bujanda at LUIS.BUJANDAFERNANDEZDEPIEROLA@osakidetza.eus). The 1000 Genomes phase 3 dataset (GRCh37) is available in PLINK2 binary format at PLINK 2.0 Resources(https://www.cog-genomics.org/plink/2.0/resources#1kg_phase3). The PRS weight files generated by this study are available in PGS catalog (https://www.pgscatalog.org/) with accession number: PGS003852.

## Code availability
All data and statistical analysis tools used in the present study are open source, details of which are available in Methods and Nature Portfolio Reporting Summary. No customized code was used to process or analyze data.

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

## Acknowledgements

National Cancer Institute, National Human Genome Research Institute (R01 CA244588 (Ulrike Peters), U01 CA164930 (Ulrike Peters), U01 CA137088 (Ulrike Peters), R01 CA059045 (Ulrike Peters), R01 CA201407 (Ulrike Peters), R01 CA206279 (Ulrike Peters), U01 CA261339 (David V Conti), R01 CA185094 (Ulrike Peters), U01 HG008657 (Gail P. Jarvik)).

## Author contributions

M.T., S.L.S., L.L.M., M.A.J., S.T.V., G.P.J., U.P., and L.H. designed the study. F.J.BvD., A.S., A.B.H., A.G., D.D.B., D.V.C., D.H.K., E.W., H.H., H.B., J.C.C., J.K.,J.H., L.L.M., L.B., M.A.J., M.D.A., M.L.W., M.S., M.L.D., R.Pe., R.E.S., R.W.H., S.Kü., S.C.B., S.T., S.B., T.A.H., V.V., V.,A., J.D.P., C.I.L., J.C.F., I.L.V., P.D.P.P., R.S.H., G.P.J., I.P.T., W.Z., D.A.C., U.P., and L.H., recruited patients and collected samples. M.T., Y.S., E.A.R., L.C.S., S.L.S., M.N.T., Z.C., C.F.R., P.L., N.M., R.C.T., V.D.O., S.J., A.W., A.I.P., A.T.C., A.G.Z., A.H.W., A.L., C.Y.U., C.M.T., C.G., C.N., C.A.H., C.Q., D.T.B., D.D.B., D.R.C., D.V.C., D.H.K., E.H., E.S., F.R.S., G.R., G.G.G., H.H., I.O., J.H.O., J.K.L., J.L.S., J.C.C., J.K., J.R.H., J.Z., J.G., J.L.H., J.R.P., K.V., Ke. M., Ko. M., K.J.J., L.L., L.L.M., L.V., L.B., M.J.G., M.M., M.L.S., M.D.A., M.L.W., M.H., M.O.W., M.S.K., M.S., M.I., M.L.D., N.U., N.S., P.V., P.T.C., P.A.N., Q.C., R.Pe., R.Pa., R.E.S., R.S.S., R.W.H., R.V., R.Pr., S.Kü., S.C.B., S.T., S.I.B., L.S.C., S.B., S.J.W., S.J.C., S.H.J., S.Kw., T.A.H., T.Y., T.O.K., V.V., V.A., W.J., X.S., Y.L., Y.A., Z.K.S., B.V.G., C.M.U., E.A.P., J.D.P., C.I.L., R.M., V.M., J.C.F., G.C., I.L.V., M.G.D., S.B.G., R.B.H., P.D.P.P., R.S.H., G.P.J., I.P.T., W.Z., D.A.C., U.P., and L.H. analyzed and interpreted the data. All authors drafted or substantially revised the manuscript. L.H. and U.P. supervised the study and acquired funding, are corresponding authors.

## Competing interests

D.A.C. receives funds from NCI. K.V. receives related Research support from Cepheid and non-financial collaboration with Optra Health. L.B. is a consultant or has received research funding from Ikan Biotech. R.E.S. got research support from Immunovia, Freenome, Exact Sciences. Z.K.S.'s immediate family member serves as a consultant in Ophthalmology for Adverum, Genentech, Gyroscope Therapeutics Limited, Neurogene, Optos Plc, Outlook Therapeutics, RegenexBio, and Regeneron (outside the submitted work). The remaining authors declare no competing interests.

## Additional information

Minta Thomas [1], Yu-Ru Su[1,2], Elisabeth A. Rosenthal [3], Lori C. Sakoda [1,4], Stephanie L. Schmit [5,6], Maria N. Timofeeva [7,8], Zhishan Chen[9], Ceres Fernandez-Rozadilla [10,11], Philip J. Law [12], Neil Murphy [13], Robert Carreras-Torres[14], Virginia Diez-Obrero[15,16,17], Franzel J. B. van Duijnhoven[18], Shangqing Jiang[1], Aesun Shin [19], Alicja Wolk [20], Amanda I. Phipps[1,21], Andrea Burnett-Hartman[22], Andrea Gsur [23], Andrew T. Chan [24,25,26,27,28,29], Ann G. Zauber [30], Anna H. Wu[31], Annika Lindblom[32,33], Caroline Y. Um [34], Catherine M. Tangen[35], Chris Gignoux [36], Christina Newton[34], Christopher A. Haiman[37], Conghui Qu[1], D. Timothy Bishop [38], Daniel D. Buchanan [39,40,41], David R. Crosslin[42], David V. Conti[37], Dong-Hyun Kim[43], Elizabeth Hauser[44,45], Emily White[1,46], Erin Siegel [47], Fredrick R. Schumacher [48], Gad Rennert[49,50], Graham G. Giles [51], Heather Hampel [52], Hermann Brenner [53,54], Isao Oze [55], Jae Hwan Oh[56], Jeffrey K. Lee[57,58], Jennifer L. Schneider[4], Jenny Chang-Claude [59,60], Jeongseon Kim [61], Jeroen R. Huyghe [1], Jiayin Zheng [1], Jochen Hampe [62], Joel Greenson[58], John L. Hopper[63,64], Julie R. Palmer[65], Kala Visvanathan [66], Keitaro Matsuo [67], Koichi Matsuda [68], Keum Ji Jung [69], Li Li[70], Loic Le Marchand[71], Ludmila Vodickova[72,73,74], Luis Bujanda [75], Marc J. Gunter[13], Marco Matejcic[76], Mark A. Jenkins [40,63], Martha L. Slattery[77], Mauro D'Amato[78,79], Meilin Wang [80], Michael Hoffmeister [53], Michael O. Woods[81], Michelle Kim[1],

Mingyang Song [24,26,82], Motoki Iwasaki[83,84], Mulong Du [85,86], Natalia Udaltsova[4], Norie Sawada [84], Pavel Vodicka[72,73], Peter T. Campbell[87], Polly A. Newcomb [1], Qiuyin Cai [9], Rachel Pearlman[52], Rish K. Pai[88], Robert E. Schoen [89], Robert S. Steinfelder[1], Robert W. Haile [90], Rosita Vandenputtelaar[91], Ross L. Prentice[1], Sébastien Küry [92], Sergi Castellví-Bel [93], Shoichiro Tsugane [84], Sonja I. Berndt[94], Soo Chin Lee[95], Stefanie Brezina [23], Stephanie J. Weinstein [94], Stephen J. Chanock [94], Sun Ha Jee[96], Sun-Seog Kweon[97,98], Susan Vadaparampil[82], Tabitha A. Harrison [1], Taiki Yamaji[83], Temitope O. Keku[99], Veronika Vymetalkova[72,73], Volker Arndt [53], Wei-Hua Jia [100], Xiao-Ou Shu[101], Yi Lin[1], Yoon-Ok Ahn[19], Zsofia K. Stadler [102], Bethany Van Guelpen [103,104], Cornelia M. Ulrich [105], Elizabeth A. Platz [66], John D. Potter [1], Christopher I. Li[1], Reinier Meester[91], Victor Moreno [106,107,108,109], Jane C. Figueiredo[37,110], Graham Casey[111], Iris Lansdorp Vogelaar[91], Malcolm G. Dunlop [8], Stephen B. Gruber [112], Richard B. Hayes [113], Paul D. P. Pharoah [114], Richard S. Houlston [12], Gail P. Jarvik[3], Ian P. Tomlinson [11], Wei Zheng [9], Douglas A. Corley[4,115], Ulrike Peters [1,21] ✉ & Li Hsu [1,116] ✉

[1]Public Health Sciences Division, Fred Hutchinson Cancer Center, Seattle, WA 98109, USA. [2]Biostatistics Division, Kaiser Permanente Washington Health Research Institute, Seattle, USA. [3]Department of Medicine (Medical Genetics), University of Washington Medical Center, Seattle, WA 98195, USA. [4]Division of Research, Kaiser Permanente Northern California, Oakland, CA, USA. [5]Genomic Medicine Institute, Cleveland Clinic, Cleveland, OH, USA. [6]Population and Cancer Prevention Program, Case Comprehensive Cancer Center, Cleveland, USA. [7]Danish Institute for Advanced Study (DIAS), Epidemiology, Biostatistics and Biodemography, Department of Public Health, University of Southern Denmark, Odense, Denmark. [8]Colon Cancer Genetics Group, Medical Research Council Human Genetics Unit, Institute of Genetics and Cancer, University of Edinburgh, Edinburgh EH4 2XU, U, Germany. [9]Division of Epidemiology, Department of Medicine, Vanderbilt Epidemiology Center, Vanderbilt University Medical Center, Nashville, TN, USA. [10]Instituto de Investigacion Sanitaria de Santiago (IDIS), Choupana sn, 15706 Santiago de Compostela, Spain. [11]Edinburgh Cancer Research Centre, Institute of Genomics and Cancer, University of Edinburgh, Crewe Road, Edinburgh EH4 2XU, UK. [12]Division of Genetics and Epidemiology, The Institute of Cancer Reseach, London SW7 3RP, UK. [13]Nutrition and Metabolism Branch, International Agency for Research on Cancer, World Health Organization, Lyon, France. [14]Digestive Diseases and Microbiota Group, Girona Biomedical Research Institute (IDIBGI), Salt 17190 Girona, Spain. [15]Unit of Biomarkers and Susceptibility, Oncology Data Analytics Program, Catalan Institute of Oncology, Barcelona 08908, Spain. [16]Colorectal Cancer Group, ONCOBELL Program, Bellvitge Biomedical Research Institute, Barcelona 08908, Spain. [17]Department of Clinical Sciences, Faculty of Medicine, University of Barcelona, Barcelona 08908, Spain. [18]Division of Human Nutrition and Health, Wageningen University & Research, Wageningen, The Netherlands. [19]Department of Preventive Medicine, Seoul National University College of Medicine, Seoul National University Cancer Research Institute, Seoul, South Korea. [20]Institute of Environmental Medicine, Karolinska Institutet, Stockholm, Sweden. [21]Department of Epidemiology, University of Washington, Seattle, WA, USA. [22]Institute for Health Research, Kaiser Permanente Colorado, Denver, CO, USA. [23].Center for Cancer Research, Medical University Vienna, Vienna, Austria. [24]Division of Gastroenterology, Massachusetts General Hospital and Harvard Medical School, Boston, MA, USA. [25]Channing Division of Network Medicine, Brigham and Women's Hospital and Harvard Medical School, Boston, MA, USA. [26]Clinical and Translational Epidemiology Unit, Massachusetts General Hospital and Harvard Medical School, Boston, MA, USA. [27]Broad Institute of Harvard and MIT, Cambridge, MA, USA. [28]Department of Epidemiology, Harvard T.H. Chan School of Public Health, Harvard University, Boston, MA, USA. [29]Department of Immunology and Infectious Diseases, Harvard T.H. Chan School of Public Health, Harvard University, Boston, MA, USA. [30]Department of Epidemiology and Biostatistics, Memorial Sloan Kettering Cancer Center, New York, NY, USA. [31]University of Southern California, Preventative Medicine, Los Angeles, CA, USA. [32]Department of Clinical Genetics, Karolinska University Hospital, Stockholm, Sweden. [33]Department of Molecular Medicine and Surgery, Karolinska Institutet, Stockholm, Sweden. [34]Department of Population Science, American Cancer Society, Atlanta, GA, USA. [35]SWOG Statistical Center, Fred Hutchinson Cancer Research Center, Seattle, WA, USA. [36]Colorado Center for Personalized Medicine, University of Colorado - Anschutz Medical Campus, Aurora, CO, USA. [37]Department of Population and Public Health Sciences, Keck School of Medicine, University of Southern California, Los Angeles, CA, USA. [38]Leeds Institute of Cancer and Pathology, University of Leeds, Leeds, UK. [39]Colorectal Oncogenomics Group, Department of Clinical Pathology, The University of Melbourne, Parkville, VIC 3000, Australia. [40]University of Melbourne Centre for Cancer Research, Victorian Comprehensive Cancer Centre, Parkville, VIC 3000, Australia. [41]Genomic Medicine and Family Cancer Clinic, The Royal Melbourne Hospital, Parkville, VIC 3000, Australia. [42]Department of Bioinformatics and Medical Education, University of Washington Medical Center, Seattle, WA 98195, USA. [43]Department of Social and Preventive Medicine, Hallym University College of Medicine, Okcheon-dong, South Korea. [44]VA Cooperative Studies Program Epidemiology Center, Durham Veterans Affairs Health Care System, Durham, NC, USA. [45]Duke Molecular Physiology Institute, Duke University Medical Center, Durham, NC, USA. [46]Department of Epidemiology, University of Washington School of Public Health, Seattle, WA, USA. [47]Cancer Epidemiology Program, H. Lee Moffitt Cancer Center and Research Institute, Tampa, FL, USA. [48]Department of Population and Quantitative Health Sciences, Case Western Reserve University, Cleveland, OH, USA. [49]Department of Community Medicine and Epidemiology, Lady Davis Carmel Medical Center, Haifa, Israel. [50]Ruth and Bruce Rappaport Faculty of Medicine, Technion-Israel Institute of Technology, Haifa, Israel. [51]Cancer Epidemiology Division, Cancer Council Victoria, Melbourne, VIC, Australia. [52]Division of Human Genetics, Department of Internal Medicine, The Ohio State University Comprehensive Cancer Center, Columbus, OH, USA. [53]Division of Clinical Epidemiology and Aging Research, German Cancer Research Center (DKFZ), Heidelberg, Germany. [54]Division of Preventive Oncology, German Cancer Research Center (DKFZ) and National Center for Tumor Diseases (NCT), Heidelberg, Germany. [55].Division of Cancer Epidemiology and Prevention, Aichi Cancer Center Research Institute, Nagoya, Japan. [56].Research Institute and Hospital, National Cancer Center, Goyang, South Korea, South Korea. [57].Department of Gastroenterology, Kaiser Permanente San Francisco Medical Center, San Francisco, CA, USA. [58]Department of Pathology, University of Michigan, Ann Arbor, MI 48104, USA. [59]Division of Cancer Epidemiology, German Cancer Research Center (DKFZ), Heidelberg, Germany. [60]University Medical Centre Hamburg-Eppendorf, University Cancer Centre Hamburg (UCCH), Hamburg, Germany. [61]Department of Cancer Biomedical Science, Graduate School of Cancer Science and Policy, National Cancer Center, Gyeonggi-do, South Korea. [62]Department of Medicine I, University Hospital Dresden, Technische Universität Dresden (TU Dresden), Dresden, Germany. [63]Centre for Epidemiology and Biostatistics, Melbourne School of Population and Global Health, The University of Melbourne, Melbourne, VIC, Australia. [64]Department of Epidemiology, School of Public Health and Institute of Health and Environment, Seoul National University, Seoul, South Korea. [65]Slone Epidemiology Center, School of Medicine, Boston University, Boston, MA, USA. [66]Department of Epidemiology, Johns Hopkins Bloomberg School of Public Health, Baltimore, MD, USA. [67]Division of Molecular and Clinical Epidemiology, Aichi Cancer Center Research Institute, Nagoya, Japan. [68]Laboratory of Clinical Genome Sequencing, Department of Computational Biology and Medical Sciences, Graduate School of Frontier Sciences, University of Tokyo, Tokyo, Japan. [69]Institute for Health Promotion, Graduate School of Public Health, Yonsei University, Seoul, Korea. [70]Department

of Family Medicine, University of Virginia, Charlottesville, VA, USA. [71]University of Hawaii Cancer Center, Honolulu, HI, USA. [72]Department of Molecular Biology of Cancer, Institute of Experimental Medicine of the Czech Academy of Sciences, Prague, Czech Republic. [73]Institute of Biology and Medical Genetics, First Faculty of Medicine, Charles University, Prague, Czech Republic. [74]Faculty of Medicine and Biomedical Center in Pilsen, Charles University, Pilsen, Czech Republic. [75]Department of Gastroenterology, Biodonostia Health Research Institute, Centro de Investigación Biomédica en Red de Enfermedades Hepáticas y Digestivas (CIBERehd), Universidad del País Vasco (UPV/EHU), San Sebastián, Spain. [76]Moffitt Cancer Center, Tampa, FL, USA. [77]Department of Internal Medicine, University of Utah, Salt Lake City, UT, USA. [78]Department of Medicine and Surgery, LUM University, Camassima, Italy. [79]Gastrointestinal Genetics Lab, CIC bioGUNE-BRTA, Derio, Spain. [80]Department of Environmental Genomics, School of Public Health, Nanjing Medical University, Nanjing, China. [81]Memorial University of Newfoundland, Discipline of Genetics, St. John's, Canada. [82]Departments of Epidemiology and Nutrition, Harvard TH Chan School of Public Health, Boston, MA, USA. [83]Division of Epidemiology, National Cancer Center Institute for Cancer Control, National Cancer Center, Tokyo, Japan. [84]Division of Cohort Research, National Cancer Center Institute for Cancer Control, National Cancer Center, Tokyo, Japan. [85]Department of Biostatistics, School of Public Health, Nanjing Medical University, Nanjing, China. [86]Department of Environmental Health, Harvard T.H. Chan School of Public Health, Boston, MA, USA. [87]Department of Epidemiology and Population Health, Albert Einstein College of Medicine, Bronx, NY, USA. [88]Department of Laboratory Medicine and Pathology, Mayo Clinic Arizona, Scottsdale, AZ, USA. [89]Department of Medicine and Epidemiology, University of Pittsburgh Medical Center, Pittsburgh, PA, USA. [90]Samuel Oschin Comprehensive Cancer Institute, CEDARS-SINAI, Los Angeles, CA, USA. [91]Department of Public Health, Erasmus University Medical Center, Rotterdam, The Netherlands. [92]Nantes Université, CHU Nantes, Service de Génétique Médicale, F-44000 Nantes, France. [93]Gastroenterology Department, Hospital Clínic, Institut d'Investigacions Biomèdiques August Pi i Sunyer (IDIBAPS), Centro de Investigación Biomédica en Red de Enfermedades Hepáticas y Digestivas (CIBEREHD), University of Barcelona, Barcelona, Spain. [94]Division of Cancer Epidemiology and Genetics, National Cancer Institute, National Institutes of Health, Bethesda, MD, USA. [95]National University Cancer Institute, Singapore, Singapore. [96]Department of Epidemiology and Health Promotion, Graduate School of Public Health, Yonsei University, Seoul, Korea. [97]Department of Preventive Medicine, Chonnam National University Medical School, Gwangju, Korea. [98]Jeonnam Regional Cancer Center, Chonnam National University Hwasun Hospital, Hwasun, Korea. [99]Center for Gastrointestinal Biology and Disease, University of North Carolina, Chapel Hill, NC, USA. [100]State Key Laboratory of Oncology in South China, Cancer Center, Sun Yat-sen University, Guangzhou, China. [101]Vanderbilt University Medical Center, Nashville, TN, USA. [102]Department of Medicine, Memorial Sloan Kettering Cancer Center, New York, NY, USA. [103]Department of Radiation Sciences, Oncology Unit, Umeå University, Umeå, Sweden. [104]Wallenberg Centre for Molecular Medicine, Umeå University, Umeå, Sweden. [105]Huntsman Cancer Institute and Department of Population Health Sciences, University of Utah, Salt Lake City, UT, USA. [106]Oncology Data Analytics Program, Catalan Institute of Oncology-IDIBELL, L'Hospitalet de Llobregat, Barcelona, Spain. [107]CIBER Epidemiología y Salud Pública (CIBERESP), Madrid, Spain. [108]Department of Clinical Sciences, Faculty of Medicine, University of Barcelona, Barcelona, Spain. [109]ONCOBEL Program, Bellvitge Biomedical Research Institute (IDIBELL), L'Hospitalet de Llobregat, Barcelona, Spain. [110]Department of Medicine, Samuel Oschin Comprehensive Cancer Institute, Cedars-Sinai Medical Center, Los Angeles, CA, USA. [111]Center for Public Health Genomics, University of Virginia, Charlottesville, VA, USA. [112]Department of Medical Oncology & Therapeutics Research, City of Hope National Medical Center, Duarte, CA, USA. [113]Division of Epidemiology, Department of Population Health, New York University School of Medicine, New York, NY, USA. [114]Department of Public Health and Primary Care, University of Cambridge, Cambridge, UK. [115]Department of Gastroenterology, Kaiser Permanente Medical Center, San Francisco, CA, USA. [116]Department of Biostatistics, University of Washington, Seattle, WA, USA. ✉e-mail: upeters@fredhutch.org; lih@fredhutch.org

