## [Peer Review File · Nature Communications]

Combining Asian and European Genome-Wide Association Studies of Colorectal Cancer Improves Risk Prediction Across Racial and Ethnic PopulationsREVIEWER COMMENTS

Reviewer #1 (Remarks to the Author):

The main contribution of this manuscript is to provide an improved cross-ancestral PRS for colorectal cancer risk assessment for use across four major racial/ethnic groups in the US. This was done by combining Asian and European ancestry GWAS data; and evaluating discrimination/calibration of the PRS in four racial/ethnic groups in independent datasets from multiple studies. The improvement in the PRS is modest, and performance is particularly poor for individuals of African ancestry. However, this well-conducted study is still an important contribution to the literature, until additional studies including larger numbers of more diverse populations are available for PRS development and validation.

My main comments are on the analyses to evaluate the potential clinical utility of PRS. The authors used a net benefit analyses in the GERA cohort to show that for a risk threshold used currently to identify individuals eligible for colorectal cancer screening in the US (>45 years, corresponding to 10-year risk>0.26%), PRS improves modestly the net benefit compared to using only information on family history. Although a full assessment of clinical utility is clearly outside the scope of this paper, additional analyses using data at hand would provide a more intuitive and informative assessment of the potential clinical utility:

1. In addition to the net benefit curve, show estimates of number of high-risk subjects with and without colorectal cancer and true+/false+ proportions, for different risk thresholds (e.g. Fig 3 in Kerr et al. JCO 2016 <https://www.ncbi.nlm.nih.gov/pmc/articles/PMC4962736/>)
2. Given that clinical decisions are based on absolute risk, estimation of the absolute risk stratification in the four US population groups considered could be added using RR's estimates and PRS distributions estimated in this paper, combined with population-based incidence and competing mortality rates (e.g. from CDC and SEER). This will give the reader a better sense of difference in absolute risk for individuals identified at high risk (e.g. top 10 or 5%) using family history or family history+PRS risk scores.

The additional analyses suggested above will give a better sense of the potential clinical utility of PRS in colorectal cancer screening. In particular, it would be informative to see how a population can be stratified into different levels of risk (not just a single threshold), as the

authors point out in the discussion that "the purpose of PRS is not to identify CRS, but rather stratify individuals into different risk strata".

This reference evaluating potential clinical utility of colorectal PRS in UKBiobank seems relevant for the discussion: Biggs et al BMJ 2022 <https://www.bmj.com/content/379/bmj-2022-071707>

Reviewer #2 (Remarks to the Author):

In this paper, the authors added Asian ancestry data into the European ancestry training set, and reported improved AUC for Asian, Black, Hispanic populations, with AUC ranging from 0.59-0.65. The authors also compared 2 approaches based on ancestry-specific PRS vs single cross-ancestry Asian-European PRS, and reported that the 2 approaches performed similarly for Asians and non-Hispanic European, and cross-ancestry approach improved risk prediction for Blacks and Hispanics.

Comments:

1. The improved AUC remained relatively modest, as 0.63 for Asian population, 0.65 for non-Hispanic White, and 0.59 for Blacks. Does these models include clinical factors such as family history, demographics, smoking, and more known CRC risk factors? It would be helpful to have a better description of what is included in the risk model itself.
2. Some of the AUC improvement is rather modest. For example 0.58 to 0.59 for the Black population. It is unclear if this work provides added values to these racial groups that do not have ancestry-specific GWAS available, such as Blacks, Hispanics, etc.
3. Figure 3, which dataset was used to assess the calibration? This is not specified.
4. The clinical decision curve analysis is potentially useful. However the authors did not expand on the interpretations. How would the model affect the intervention under the range of threshold probability. What would be considered as a reasonable threshold probability for CRC?

What would be the corresponding net benefit?

5. While PRS-CSx and LDPred2 are well-used packages, it is important for the readership to include some brief descriptions of the methodology used in these packages.

REVIEWER COMMENTS

Reviewer #1 (Remarks to the Author):

The main contribution of this manuscript is to provide an improved cross-ancestral PRS for colorectal cancer risk assessment for use across four major racial/ethnic groups in the US. This was done by combining Asian and European ancestry GWAS data; and evaluating discrimination/calibration of the PRS in four racial/ethnic groups in independent datasets from multiple studies. The improvement in the PRS is modest, and performance is particularly poor for individuals of African ancestry. However, this well-conducted study is still an important contribution to the literature, until additional studies including larger numbers of more diverse populations are available for PRS development and validation.

My main comments are on the analyses to evaluate the potential clinical utility of PRS. The authors used a net benefit analysis in the GERA cohort to show that for a risk threshold used currently to identify individuals eligible for colorectal cancer screening in the US (>45 years, corresponding to 10-year risk>0.26%), PRS improves modestly the net benefit compared to using only information on family history. Although a full assessment of clinical utility is clearly outside the scope of this paper, additional analyses using data at hand would provide a more intuitive and informative assessment of the potential clinical utility:

1. In addition to the net benefit curve, show estimates of number of high-risk subjects with and without colorectal cancer and true+/false+ proportions, for different risk thresholds (e.g. Fig 3 in Kerr et al. JCO 2016 <https://www.ncbi.nlm.nih.gov/pmc/articles/PMC4962736/>)

Response:

Following this helpful suggestion, we included the estimates of the number of high-risk individuals and how many of these would be true positives (cases), and true- and false- positive rates for different risk thresholds based on the prediction model as part of the revised integrated Figure 4 (we have expanded Figure 4 to additionally include three subfigures for the family history (FamHx) only and FamHx+PRS models, true- and false- positive rates (Figure 4(b)), number of high-risk individuals at different risk threshold (Figure 4(c)) and number of high-risk individuals with CRC at different risk threshold (Figure 4(d)), provided below.

To address this comment, we added the following text in the manuscript, page 34, paragraph 2.

Specifically, let r_t be the risk threshold and $I(t)$ the cumulative incidence of developing CRC for an individual by time t in the presence of competing risks, here, death¹. Further, we define $z = 1$ to indicate that an individual is at high risk if their predicted t -year risk from the model is greater than or equal to r_t and $z = 0$ otherwise. We chose the landmark time $t = 10$ years. At each r_t , we calculated the number of true and false positives, TP_{r_t} and FP_{r_t} , by $I(t|z=1) \times P(z=1) \times N$ and $\{1 - I(t|z=1)\} \times P(z=1) \times N$, respectively, where N is the total number of participants. The true-positive rate was then calculated as $TP_{r_t}/TP_{r_t=0}$ and the false-positive rate was calculated as $FP_{r_t}/FP_{r_t=0}$.

We added the following text in the manuscript, page number 23, 1st paragraph.

For example, in the GERA cohort, using the risk threshold 0.29%, which corresponds to the average 10-year risk of developing CRC for a 45-year old, about 8,472 of 22,628 individuals with age 40-49 were deemed to be at high risk based on our model of family history and PRS. Among these, 99 developed CRC in the next 10 years. For this age group, a total of 149 individuals developed CRC. The true-positive and false-positive rates were 70% and 37%, respectively. Whereas, for the model based on family history only, at the same risk threshold, about 2357 would be deemed at high risk, and 37 developed CRC. The true-positive and false-positive rates were 31% and 10%, respectively.

Figure 4 (a) standardized net benefit for none, all, family history (FamHx) model, and FamHx+PRS model, and for the FamHx and FamHx+PRS models (b) true- and false-positive rates, (c) number of high-risk, and (d) number of high risk participants developed CRC at different risk thresholds, in 22,628 participants aged 40-49 from the GERA cohort.

Supplemental Table 4: Net benefit (NB) of intervention (e.g., screening) for all participants according to the proposed family history (FamHx) + PRS model and the FamHx only model for a given risk threshold.

Risk threshold (%)	NB			Advantage of Model Compared to Treat All			
	Treat All	FamHx Model	FamHx + PRS Model	FamHx Model		FamHx + PRS Model	
				Net benefit	Reduction of False Positive per 100	Net benefit	Reduction of False Positive per 100
0.04	0.002763	0.002763	0.002766	0	0	3.05E-06	1
0.08	0.002364	0.002364	0.002378	0	0	1.40E-05	2
0.09	0.002264	0.002264	0.002307	0	0	4.33E-05	5
0.11	0.002064	0.002064	0.002151	0	0	8.66E-05	8
0.13	0.001864	0.001864	0.001976	0	0	0.000112	9
0.14	0.001764	0.001764	0.001945	0	0	0.00018	13
0.16	0.001564	0.001564	0.001809	0	0	0.000245	15
0.17	0.001464	0.001464	0.001751	0	0	0.000286	17
0.19	0.001264	0.001264	0.001683	0	0	0.000419	22
0.20	0.001164	0.001164	0.001678	0	0	0.000513	26
0.22	0.000964	0.000964	0.001545	0	0	0.000581	26
0.23	0.000864	0.000864	0.00141	0	0	0.000547	24
0.25	0.000663	0.000663	0.00142	0	0	0.000757	30
0.26	0.000563	0.000563	0.001285	0	0	0.000722	28
0.28	0.000363	0.000707	0.001222	0.000344	12	0.000859	31
0.29	0.000263	0.000696	0.001142	0.000434	15	0.00088	30
0.31	6.20E-05	0.000676	0.001084	0.000613	20	0.001022	33
0.32	-3.83E-05	0.000665	0.001006	0.000703	22	0.001045	33

In addition, we added *Supplemental Table 4* to describe the clinical utility of our model in simple, clinically terms based on the decision curve analysis².

On page 23, paragraph 2 of the manuscript, we also added the following text:

Supplemental Table 4 provides more detailed results of the net benefit (NB) analysis for our proposed family history and PRS-based model and the family history-based model compared to treat all for risk thresholds (%) from 0 to 0.32%, where NB for treat all become negative. Using the same risk threshold 0.29% as in the previous example, the NB of our model is 0.11%. This can be interpreted as that compared with assuming that all individuals do not have intervention, our model with 0.11% NB leads to the equivalent of a net 11 true-positives per 10,000 individuals without an increase in the number of false-positives. Moreover, the net benefit for the model was 0.08% greater than assuming all individuals had intervention and 0.04% greater than family history-based model. We also calculated the reduction in the number of false positives per 100 patients as: $(\text{net benefit of the model} - \text{net benefit of treat all}) / (1 - \text{rt}) \times 100$. There were 30 fewer false-positives per 100 individuals for our whereas there were only 15 fewer false-positives for the family history-based model.

2. Given that clinical decisions are based on absolute risk, estimation of the absolute risk stratification in the four US population groups considered could be added using RR's estimates and PRS distributions estimated in this paper, combined with population-based incidence and competing mortality rates (e.g. from CDC and SEER). This will give the reader a better sense of difference in absolute risk for individuals identified at high risk (e.g. top 10 or 5%) using family history or family history+PRS risk scores.

The additional analyses suggested above will give a better sense of the potential clinical utility of PRS in colorectal cancer screening. In particular, it would be informative to see how a population can be stratified into different levels of risk (not just a single threshold), as the authors point out in the discussion that "the purpose of PRS is not to identify CRS, but rather stratify individuals into different risk strata".

This reference evaluating potential clinical utility of colorectal PRS in UKBiobank seems relevant for the discussion: Biggs et al BMJ 2022 <https://www.bmj.com/content/379/bmj-2022-071707>

Response:

We thank the reviewer for the valuable suggestion. Since one of independent validation datasets, GERA, is a large-scale community-based cohort study, we can directly examine the disease incidence from the data, as shown in the following Kaplan-Meier plots allowing us to empirically assess the performance of the model. Accordingly, we did not need to use external population-based incidence and competing mortality rates (e.g., SEER registry), as we and others have previously done^{3,4}. We added the Kaplan-Meier curves (Supplemental Figures 8 and 9), stratified by family history status and by quantiles of PRS with top 5%, top 25%, 25%-75%, bottom 25% and bottom 5%, for different racial/ethnic groups of GERA participants.

Supplemental Figure 8: Kaplan-Meier cumulative incidence curves of a) Family history positive (FamHx+) and b) FamHx- for different racial and ethnic groups.

Supplemental Figure 9: Kaplan-Meier cumulative incidence curves by quantiles of PRS, top 5%, top 25%, 25-75%, bottom 25%, and bottom 5% for a) African American, b) Asian, c) Hispanic, and d) Non-Hispanic White (NHW) groups.

There was clear separation between those who were in bottom and top PRS quantiles across ancestral groups, except for the African American group where the separation is less obvious due to the lower

performance and very limited number of CRC cases in this group. The probabilities of developing CRC by age 70 for top 5% of PRS ranged from 2.2 % to 4.7%, across the four different racial and ethnic groups. In comparison, the probabilities of developing CRC for those who had the positive family history were 1.9% to 5%.

We have added the details in the *manuscript page 24*. Figures are added to the Supplemental text, as Supplemental *Figures 8 and 9 (Supplemental Text Pages 20-21)*

Reviewer #2 (Remarks to the Author):

In this paper, the authors added Asian ancestry data into the European ancestry training set, and reported improved AUC for Asian, Black, Hispanic populations, with AUC ranging from 0.59-0.65. The authors also compared 2 approaches based on ancestry-specific PRS vs single cross-ancestry Asian-European PRS, and reported that the 2 approaches performed similarly for Asians and non-Hispanic European, and cross-ancestry approach improved risk prediction for Blacks and Hispanics.

Comments:

1. The improved AUC remained relatively modest, as 0.63 for Asian population, 0.65 for non-Hispanic White, and 0.59 for Blacks. Do these models include clinical factors such as family history, demographics, smoking, and more known CRC risk factors? It would be helpful to have a better description of what is included in the risk model itself.

Response:

For clarity, we added the following text in *the methods section on page 31*.

“Our model was focused on only PRS development and did not include any lifestyle and environmental risk factors.”

In the discussion on *page 28*, we wrote “Combining PRS and other CRC-associated risk factors such as lifestyle/environmental risk factors and high penetrance genes will likely further improve the prediction ”

2. Some of the AUC improvement is rather modest. For example, 0.58 to 0.59 for the Black population. It is unclear if this work provides added values to these racial groups that do not have ancestry-specific GWAS available, such as Blacks, Hispanics, etc.

Response:

We agree that we have not included African American and Hispanics GWAS data in our training data sets; however, our study demonstrates that despite lack of GWAS data in these groups, including samples from Asian and more samples from non-Hispanic White people have improved the performance in Hispanic and, to a smaller extent, in Black/African American individuals.

In the discussion, *on page 25*, we revised "... the cross-ancestry Asian-European PRS also improved risk prediction performance in Hispanic individuals and, to a smaller extent, in Black/African American individuals."

3. Figure 3, which dataset was used to assess the calibration? This is not specified.

Response:

The model-based relative risk calibration as shown in Figure 3 were based on the validation studies listed in Table 1.

For clarification, we added the following *text on page 32*, 'We evaluated the model performance using a wide range metrics, the Area Under the Receive Operating Characteristics curve, ancestry adjustment of PRS distribution, odds ratio estimates, and relative risk calibration based on all of the validation datasets listed in Table 1. The decision curve analysis is based on the GERA study, which was the only cohort study among our independent validation datasets '.

4. The clinical decision curve analysis is potentially useful. However, the authors did not expand on the interpretations. How would the model affect the intervention under the range of threshold probability. What would be considered as a reasonable threshold probability for CRC? What would be the corresponding net benefit?

Response:

We very much appreciate the reviewer's suggestions and comments. In decision curve analysis, we assumed the decision in question was whether an individual in the general population should undergo intervention (e.g., colonoscopy procedure), based on their risk. Overall, the model with the highest (standardized) net benefit is considered the "best" strategy in decision curve analysis. However, as argued in Kerr et al.⁵, decision curves cannot be used to choose a risk threshold (rt), but it summarizes the costs and benefits of intervention of the risk model at different rt. Briggs et al.⁶ considered rt 0.5%, 1%, 1.5%, and 2% for the colorectal cancer risk during the 8-year follow-up using the UK Biobank data including participants with age 40-69 years at study entry. In their model, they included age at study entry as part of their risk score, and older age individuals tend to have greater risk than younger age.

In our manuscript, we considered the rt to be the 10-year risk at age 45 for an average-risk person, because the current CRC screening guidelines recommend that an average-risk individual starts screening at age 45 years old⁷. We estimated such risk at 0.29% based on the large-scale community-based GERA cohort across racial and ethnic groups. We also considered rt as the 10-year risk at age 50 and 55 years, estimated to be 0.39% and 0.49%, respectively. However, as pointed out by Vickers et al., 2015¹ and Briggs et al.⁶ among others, the risk threshold considered is context/patient specific and is the level at which one feels vaguely about the benefits and harms of the intervention. In informing individual patient choice, patient and clinician preference around risk may vary considerably depending on their levels of concern around cancer and the intervention.

To address this comment, we have summarized the above details in the discussion, page 26 and pages 22-24, under Section 'Clinical utility for model based on PRS and family-history'.

Specifically, we have expanded Figure 4 to additionally include three subfigures for the family history (FamHx) only and Famhx+PRS models, true- and false- positive rates (Figure 4(b)), number of high-risk

individuals at different risk threshold (Figure 4(c)) and number of high-risk individuals with CRC at different risk threshold (Figure 4(d)). Based on these figures, we observed that for example, in the GERA cohort, using the risk threshold 0.29%, which corresponds to the average 10-year risk of developing CRC for a 45-year-old, about 8,472 of 22,628 individuals with age 40-49 were deemed to be at high risk based on our model of family history and PRS. Among these, 99 developed CRC in the next 10 years. For this age group, a total of 149 individuals developed CRC. The true-positive and false-positive rates were 70% and 37%, respectively. Whereas, for the model based on family history only, at the same risk threshold, about 2357 would be deemed at high risk, and 37 developed CRC. The true-positive and false-positive rates were 31% and 10%, respectively.

We also added *Supplemental Table 4* in the supplement text to show net benefit of intervention (e.g., screening) for our prediction model of family history and PRS in comparison with intervening all individuals and the model of family history only, at different risk thresholds.

We have added the following text in the *main manuscript, page 24, paragraph 1*.

In addition, we estimated the number of unnecessary interventions avoided for individuals with age 40-49 years old, as shown in *Supplemental Figure 7 and Supplemental Table 5*. Continuing using the 0.29% threshold as an example, risk stratification based on the family history and PRS would avoid 17 more interventions per 100 individuals, compared with the model based on family history, which would avoid 13 interventions per 100 individuals compared to intervening all.

Supplemental Figure 7: Unnecessary interventions avoided per 100 individuals with age 40-49 at risk threshold 0.29%.

Supplemental Table 5. Unnecessary interventions avoided per 100 individuals with age 40-49 for different risk thresholds, 0.29%, 0.39% and 0.49% corresponding the average 10-year risk of developing CRC at ages 45, 50 and 55 years, respectively.

Risk threshold (%)	Famhx	Famhx + PRS
0.29	13	30
0.39	34	38
0.49	45	49

5. While PRS-CSx and LDpred2 are well-used packages, it is important for the readership to include some brief descriptions of the methodology used in these packages.

Response:

Following the suggestion, we included the details of PRS-CSx and LDpred2 in the Supplement Text (pages 6 and 7), as given below:

LDpred⁸ uses a Bayesian approach for SNP selection and shrinkage for PRS with a spike-and-slab prior, based on an LD matrix and GWAS summary statistics. We used the updated version LDpred2 implemented in the R package bigsnpr. LDpred has been demonstrated to provide higher predictive performance, particularly with large GWAS sample size as in our study and addresses previous instability issues. Further, the use of a larger window of 3cM (using genetic distance rather than number of bases) improves performance when causal variants are in regions with long-range LD, such as HLA regions. In particular, LDpred assumes the following model for SNP effect sizes β_j ,

$$\beta_j \sim N\left(0, \frac{h^2}{Mp}\right) \text{ with probability } p,$$

0, otherwise.

where p is the proportion of causal variants, M the number of variants and h^2 the (SNP) heritability. The parameters were estimated using Gibbs sampler.

PRS-CSx⁹ is a Python based command line tool that integrates GWAS summary statistics and external LD reference panels from multiple populations to improve cross-population polygenic prediction. PRS-CSx is an extension of the Bayesian polygenic prediction method PRS-CS with Bayesian regression and continuous shrinkage prior. PRS-CSx couples genetic effects across populations via a shared continuous shrinkage prior, enabling more accurate effect size estimation by sharing information of summary statistics between populations, while incorporating linkage disequilibrium diversity across populations. For SNP j in population k , PRS-CSx uses a continuous shrinkage prior on its effect size β_{jk} , which can be represented as global-local scale mixtures of normals:

$$\beta_{jk} \sim N\left(0, \frac{\sigma_k^2}{N_k} \psi_j\right), \quad \psi_j \sim \text{Gamma}(a, \delta_j), \quad \delta_j \sim \text{Gamma}(b, \Phi),$$

where σ_k^2 and N_k are variance parameter and the number of individuals in population k , respectively, Φ is a global shrinkage parameter shared across all SNPs that models the overall sparseness of the genetic

architecture, and ψ_j is a local, SNP-specific shrinkage parameter that is adaptive to marginal GWAS associations. By assigning a gamma–gamma hierarchical prior on ψ_j (specifically, the Strawderman–Berger prior with $a = 1$ and $b = 1/2$), the marginal prior density of β_{jk} has a sizable amount of mass near zero to impose strong shrinkage on small noisy signals, and, in the meantime, heavy Cauchy-like tails to avoid over-shrinkage of truly nonzero effects.

References

1. Vickers, A. J., Cronin, A. M., Elkin, E. B. & Gonen, M. Extensions to decision curve analysis, a novel method for evaluating diagnostic tests, prediction models and molecular markers. *BMC Med. Inform. Decis. Mak.* **8**, 53 (2008).
2. Vickers, A. J. & Elkin, E. B. Decision curve analysis: a novel method for evaluating prediction models. *Med. Decis. Making* **26**, 565–574 (2006).
3. Jeon, J. *et al.* Determining risk of colorectal cancer and starting age of screening based on lifestyle, environmental, and genetic factors. *Gastroenterology* **154**, 2152–2164.e19 (2018).
4. Cannon-Albright, L. A., Carr, S. R. & Akerley, W. Population-Based Relative Risks for Lung Cancer Based on Complete Family History of Lung Cancer. *J. Thorac. Oncol.* **14**, 1184–1191 (2019).
5. Kerr, K. F., Brown, M. D., Zhu, K. & Janes, H. Assessing the clinical impact of risk prediction models with decision curves: guidance for correct interpretation and appropriate use. *J. Clin. Oncol.* **34**, 2534–2540 (2016).
6. Briggs, S. E. W. *et al.* Integrating genome-wide polygenic risk scores and non-genetic risk to predict colorectal cancer diagnosis using UK Biobank data: population based cohort study. *BMJ* **379**, e071707 (2022).

7. US Preventive Services Task Force *et al.* Screening for colorectal cancer: US preventive services task force recommendation statement. *JAMA* **325**, 1965–1977 (2021).
8. Vilhjálmsson, B. J. *et al.* Modeling linkage disequilibrium increases accuracy of polygenic risk scores. *Am. J. Hum. Genet.* **97**, 576–592 (2015).
9. Ruan, Y. *et al.* Improving polygenic prediction in ancestrally diverse populations. *Nat. Genet.* **54**, 573–580 (2022).

REVIEWERS' COMMENTS

Reviewer #1 (Remarks to the Author):

The revised manuscript adequately addressed my comments - there was however a misunderstanding - I suggested using population-based rates to infer absolute risk, rather than using the validation cohort, GERA, since this would be more representative of the target population. In addition, estimates by race/ancestry groups based on GERA only are quite imprecise for the groups smaller in size. I prefer the population-based approach but what the authors did is also informative.

Reviewer #2 (Remarks to the Author):

The authors were responsive to the comments and the manuscript has improved with the expanded section on clinical utility and relevant interpretation.

REVIEWER COMMENTS

REVIEWERS' COMMENTS

We thank the reviewers for the encouraging comments and careful evaluation of our manuscript, which really help us to further improve our manuscript.

Reviewer #1 (Remarks to the Author):

The revised manuscript adequately addressed my comments - there was however a misunderstanding - I suggested using population-based rates to infer absolute risk, rather than using the validation cohort, GERA, since this would be more representative of the target population. In addition, estimates by race/ancestry groups based on GERA only are quite imprecise for the groups smaller in size. I prefer the population-based approach but what the authors did is also informative.

Response: *We thank the reviewer for the valuable suggestions and for agreeing that our approach to estimate absolute risk using large-scale community-based cohort study (GERA) is also informative.*

In our paper, we empirically examined the clinical utility of our PRS model in a large-scale community-based cohort study, GERA. From this data, we can also estimate the disease incidence by PRS from the data, which we provide in the Supplement. Regarding the reviewer's input, we acknowledge that estimating disease incidence by race/ancestry group in GERA may lack precision due to smaller sample size within these groups, and we could get the absolute risk estimates using the (race and ethnic specific) SEER incidence rates. However, it is important to note that the SEER-based estimates do not provide empirical evidence for the real-world performance of our PRS model, as we demonstrate in our paper through independent validation studies.

Reviewer #2 (Remarks to the Author):

The authors were responsive to the comments and the manuscript has improved with the expanded section on clinical utility and relevant interpretation.

Response: *We thank the reviewer for the comments/suggestions.*